# Dual NGS Comparative Analysis of Liquid Biopsy (LB) and Formalin-Fixed Paraffin-Embedded (FFPE) Samples of Non-Small Cell Lung Carcinoma (NSCLC)

**DOI:** 10.3390/cancers14246084

**Published:** 2022-12-10

**Authors:** Laura Buburuzan, Maria-Anca Zamfir (Irofei), Carmen Maria Ardeleanu, Alin Horatiu Muresan, Florina Vasilescu, Ariana Hudita, Marieta Costache, Bianca Galateanu, Alexandra Puscasu, Alexandru Filippi, Natalia Motas

**Affiliations:** 1Department of Molecular Biology, Onco Team Diagnostic S.A., 012244 Bucharest, Romania; 2Department of Biochemistry and Molecular Biology, University of Bucharest, 050095 Bucharest, Romania; 3Department of Medical Oncology, Fundeni Clinical Institute, 72437 Bucharest, Romania; 4Faculty of Medicine, University of Medicine and Pharmacy Carol Davila Bucharest, 050474 Bucharest, Romania; 5Clinic of Thoracic Surgery, Institute of Oncology Prof. Dr. A. Trestioreanu Bucharest, 022328 București, Romania

**Keywords:** liquid biopsy, non-small lung carcinoma, NGS, molecular diagnostic, therapy modulation

## Abstract

**Simple Summary:**

Precision oncology approaches patients in a personalized manner based on their own tumor molecular profile, which can be investigated nowadays by NGS assay. Regardless of their incidence, in many cancers, including NSCLC, tissue harvesting for analysis purposes is an issue. Therefore, alternative methods for analyzing tumor-derived genetic material, such as liquid biopsy, hold great potential in overcoming this disadvantage and opening personalization perspectives for these patients. The main aim of the current study was to distinguish the potential differences between the molecular landscapes found in the tumor tissue and in plasma samples harvested from patients with NSCLC by NGS. As a result, we validated the potential use of the Ion Torrent™ platform and technology for NGS of cfNAs in NSCLC using Oncomine™ Pan-Cancer Cell-Free Assay as a valuable tool for prospective NSCLC monitoring and therapy modulation in a dual tissue/plasma analysis setup.

**Abstract:**

Lung cancer ranks second worldwide after breast cancer and third in Europe after breast and colorectal cancers when both sexes and all ages are considered. In this context, the aim of this study was to emphasize the power of dual analysis of the molecular profile both in tumor tissue and plasma by NGS assay as a liquid biopsy approach with impact on prognosis and therapy modulation in NSCLC patients. NGS analysis was performed both from tissue biopsies and from cfNAs isolated from peripheral blood samples. Out of all 29 different mutations detectable by both NGS panels (plasma and tumor tissue), seven different variants (24.13%; EGFR L858R in two patients, KRAS G13D and Q61H and TP53 G244D, V197M, R213P, and R273H) were detected only in plasma and not in the tumor itself. These mutations were detected in seven different patients, two of them having known distant organ metastasis. Our data show that NGS analysis of cfDNA could identify actionable mutations in advanced NSCLC and, therefore, this analysis could be used to monitor the disease progression and the treatment response and even to modulate the therapy in real time.

## 1. Introduction

According to Global Cancer Incidence, Mortality and Prevalence (GLOBOCAN) estimates of incidence in 2020, recently released by International Agency for Research on Cancer, lung cancer ranks second worldwide after breast cancer and third in Europe after breast and colorectal cancers when both sexes and all ages are considered [1]. In Romania, lung cancer had in 2020 the second highest incidence after colorectal cancer and the highest mortality rate. The number of lung cancer deaths registered in Romania equals the combined number of deaths from both colorectal and breast cancer [1]. Considering the sex-related incidence of lung cancer, men represent two-thirds of the newly diagnosed patients worldwide, with a stronger difference between male and female incidence rates in Asia and Africa. Both the highest incidence and related deaths are registered in Eastern Asia [1]. Worldwide, over 80% of lung cancer cases are non-small cell lung cancer (NSCLC), attributed to tobacco smoking [2], an increasing number of cases being registered after the world wars with a peak in the 1960s when the tobacco industry dramatically developed [3]. More than 60 carcinogenic compounds that lead to DNA damage have been identified in cigarette smoke [4] and their action produces from 1000 to 10 000 mutations per cell, according to a study by Yoshida et al. in 2020 [5]. The 2020 data provided by the Global Health Observatory database of the World Health Organization confirm that a high prevalence of tobacco smoking is positively correlated with the age-standardized incidence of lung cancer with the highest visibility in countries such as Hungary or Serbia [3]. 

For many years, PCR and Sanger sequencing were the gold standard for detecting the predictive biomarkers that harbor cancer onset and progression. However, these are methods that target single genes or even single actionable mutation screening [6]. Over the years, the number and complexity of molecular markers has grown and these traditional methods proved more and more inefficient. Testing for several actionable biomarkers by repeated PCR or sequencing panels requires a greater amount of DNA isolated from the tumor sample, a greater turnaround time of the results, and cumulative growing costs. Moreover, the priority of immunohistochemistry evaluation for a precision diagnostic often ends up with a reduced quantity of tumor tissue that is not sufficient for the upcoming evaluation of all targeted biomarkers by classical methods. Introducing NGS analysis in routine practice has the great advantage of obtaining a much more detailed molecular profile specific to each patient, in a single analysis, starting from a smaller amount of DNA, with a much shorter turnaround time, fixed costs, and greater precision [7]. This technique has also the flexibility of adapting the targeted regions of interest comprised in each NGS panel, depending on the disease. NGS analysis has also the advantage of uncovering some unexpected gene variants that can benefit from targeted therapy [8]. However, in particular cases of patients with worsening conditions, when a fast result is needed, single-gene analysis by real-time PCR is a suitable option despite the more restrictive information provided. A major disadvantage of the NGS analysis is the increased turnaround time, which often exceeds in many labs the 10 days stipulated in the international guidelines [9]. Despite the obvious informative advantage of tumor tissue NGS, the analyst encounters frequently the impediment of the lack of tissue following immunohistochemistry investigation, or the pre-analytical damage of the tumor tissue. These types of situations make NGS analysis of cell-free tumor DNA isolated from liquid biopsy, a precious instrument for molecular investigations. Liquid biopsy analysis is an important tool for patients with advanced stages of NSCLC or in situations where it is not possible to obtain a biopsy [10], and has recently been considered as a complementary investigation to the tissue-based approach [11].

The main aim of the current study was to distinguish the potential differences between the molecular landscapes found in the tumor tissue and in plasma samples harvested from patients with NSCLC by NGS. As a result, we validated the potential use of the Ion Torrent™ platform and technology for NGS of cfNAs in NSCLC using Oncomine™ Pan-Cancer Cell-Free Assay as a valuable tool for prospective NSCLC monitoring and therapy modulation in a dual tissue/plasma analysis setup. 

## 2. Materials and Methods

### 2.1. Patients

A cohort of 57 patients with suspicion of non-small cell lung carcinoma (NSCLC) was included in the study between October 2020 and June 2021. Tumor biopsies together with liquid biopsies from peripheral blood were collected simultaneously from all patients before histopathological diagnosis. Following histopathology and immunohistochemistry analysis, 19 patients with other cancers or other diagnostics were initially removed from the study cohort. Another 12 different patients were excluded from the study—11 of them due to the lack of or insufficient FFPE material remained for the NGS gDNA testing and another patient due to the total necrosis of the tumor tissue that made the material incompatible with the isolation of good-quality nucleic acids. The remaining 26 patients with confirmed NSCLC were further subjected to NGS analysis in this study. This study was developed within a grant from the Romanian Ministry of Education and Research, CCCDI—UEFISCDI, project number PN-III-P2-2.1-PTE-2019-0577, and approved by the ethical committee of OncoTeam Diagnostic (approval no. 30/10.10.2019). All the patients enrolled were informed about the study and gave their written consent regarding their participation in the research. All the collected samples were harvested as a result of a medical indication and served primarily for diagnostic purposes and secondarily for the project. 

### 2.2. Immunohistochemistry Detection

Tissue biopsies were macroscopically examined, and tumor fragments were sampled according to the routine orientation practices. The formalin-fixed fragments were embedded in paraffin and cut into 4 µm sections with a Leica RM2245 Microtome (Leica Biosystems, Deer Park, IL, USA). The hematoxylin–eosin slides were examined with a Leica DM750 Microscope (Leica Biosystems, Deer Park, IL, USA). Immunohistochemistry (IHC) staining was conducted with monoclonal primary antibodies presented in Table 1.

### 2.3. Liquid Biopsy and the cfDNA NGS Panel 

Peripheral blood was collected from each patient in Cell-Free DNA BCT^®^ CE (Strek, La Vista, NE, USA), and plasma was separated by a succession of low-speed high-speed centrifugations and fr(−80 °C) until nucleic acid isolation. Cell-free nucleic acids (cfNA) were isolated using MagMAXTM Cell-Free Total Nucleic Acid Kit (Thermo Fisher Scientific^TM^, Waltham, MA, USA) and the cfDNA was quantified with Qubit^®^ 2.0 Fluorimeter (Invitrogen™, Thermo Fisher Scientific^TM^ Waltham, MA, USA). Cell-free DNA libraries were obtained starting from 20 ng cfNA using Ion Torrent^TM^ Oncomine^TM^ Pan-Cancer Cell-Free Assay (Ion Torrent^TM^, Waltham, MA, USA) following the manufacturer’s instructions and quantified by qPCR. Insertions-deletions (indels) and single nucleotide variants (SNVs) were investigated in 44 genes with a limit of detection down to 0.1% allelic frequency: AKT1, ALK, APC, AR, ARAF, BRAF, CHEK2, CTNNB1, DDR2, EGFR, ERBB2, ERBB3, ESR1, FBXW7, FGFR1, FGFR2, FGFR3, FGFR4, FLT3, GNA11, GNAQ, GNAS, HRAS, IDH1, IDH2, KIT, KRAS, MAP2K1, MAP2K2, MET, MTOR, NRAS, NTRK1, NTRK3, PDGFRA, PIK3CA, PTEN, RAF1, RET, ROS1, SF3B1, SMAD4, SMO, TP53. Gene fusions and copy number genes (CNVs) were not evaluated at the time of the study. 

### 2.4. Tumor Tissue and gDNA NGS Panel

For tumor genomic DNA (gDNA) NGS investigations, formalin-fixed paraffin-embedded (FFPE) resection specimens or tumor biopsies were used. Tissue micro-dissected areas with at least 20% tumor cells underwent gDNA isolation with RecoverAll™ Total Nucleic Acid Isolation Kit for FFPE (Invitrogen™, Thermo Fisher Scientific^TM^, Waltham, MA, USA). A quantity of 10 ng gDNA calculated upon measurement with Qubit^®^ 2.0 Fluorimeter (Invitrogen™, Thermo Fisher Scientific^TM^, Waltham, MA, USA) was used for downstream analysis of tumor-specific variants. Preparation and equalization of the libraries were performed with Oncomine Solid Tumor DNA Kit (Ion Torrent^TM^, Waltham, MA, USA). Single nucleotide variants and indels in 22 genes, with an allelic frequency of at least 5%, were considered for mutation evaluation: AKT1, ALK, BRAF, CTNNB1, DDR2, EGFR, ERBB2, ERBB4, FGFR1, FBXW7, FGFR2, FGFR3, KRAS, MAP2K1, MET, NOTCH1, NRAS, PIK3CA, PTEN, SMAD4, STK11, TP53.

### 2.5. NGS and Sequence Analysis

The templated libraries were loaded on Ion 540™ Chips (Ion Torrent, Waltham, MA, USA), (cfDNA libraries) or Ion 520™ Chips (Ion Torrent, Waltham, MA, USA), (gDNA) using an Ion Chef™ Instrument (Ion Torrent, Waltham, MA, USA) and were subjected to sequencing using an Ion GeneStudio™ S5 Plus System (Ion Torrent^TM^, Waltham, MA, USA). Cancer-related pathogenic variants in cfDNA and gDNA were analyzed using Ion Reporter™ Software and IGV software [12]. Classification of the variants was performed by consulting NCBI (National Center for Biotechnology Information) and COSMIC (Catalogue of Somatic Mutations in Cancer) databases.

### 2.6. Tumor Tissue and PCR Evaluation of EGFR Pathogenic Variants

Formalin-fixed paraffin-embedded (FFPE) resection specimens or tumor biopsies from 23 patients were used for PCR detection of pathogenic variants in the EGFR gene. Genomic DNA was isolated using the IVD Cobas^®^ DNA Sample Preparation Kit (Roche Diagnostics International, Rotkreuz, Switzerland). The status of 42 NSCLC-associated mutations in exons 18, 19, 20, and 21 of the EGFR gene was evaluated by real-time PCR using the IVD Cobas^®^ EGFR Mutation Test v2 kit (Roche Diagnostics International, Rotkreuz, Switzerland) and a Cobas z 480 analyzer (Roche Diagnostics International, Rotkreuz, Switzerland) according to the manufacturer’s instructions to validate NGS.

### 2.7. Data Analysis

Raw data were organized in Excel and GraphPad Prism 5 and Python 3 (using the matplotlib and Seaborn libraries) were used for data analysis and visualization. Oncoprint was used to visualize gene mutation frequency using the online resource available at https://www.cbioportal.org/oncoprinter (accessed on 5 November 2022). Plasma allele frequency data followed a log-normal distribution (the Kolmogorov–Smirnov normality check failed at *p* = 0.0001 *** before normalization, but passed the test after log10 normalization), thus, for visualization purposes, log values were used. As tumor allele frequency results passed the normality check, these results were used as is. The correlation between plasma and tumor parameters was assayed by Spearman’s rank correlation, Wilcoxon signed-rank test was used for column analysis, and Fisher’s exact test was used for categorical data analysis.

## 3. Results

### 3.1. NSCLC Patients Group Characteristics

From the total of 26 patients with NSCLC, 16 patients were diagnosed with adenocarcinoma (LUAD) and 10 with squamous cell carcinoma (LUSC), following histopathology and immunohistochemistry tests. The age of the enrolled patients ranged from 44 to 81 years, with a mean of 67 years. Most of the patients diagnosed with NSCLC were men (20/26, 76.92%). The group has a high homogeneity of tumor characteristics, with all the patients having cancers with stages III/IV, and most of the cancers having poorly differentiated cells (18/26, 69.23%).

The characteristics of the NSCLC group are described in Table 2.

Plasma cfDNA testing was successful for all 26 patients. The volume of plasma separated from the liquid biopsies ranged from 4 mL to 10 mL (median, 5 mL). The concentration of cfDNA isolated from plasma spanned from 2.1 ng/μL to 33.8 ng/μL (median, 4.42 ng/μL). 

### 3.2. Gene Variants Detected

A total of 34 different variants were detected in tumor and plasma samples of our 26 NSCLC patients’ group, however, five variants found in plasma were not included in the tumor analysis panel. From the remaining 29 different variants, concordance between plasma and tumor was found for 21 variants (61.76%) (Table 3). Four patients did not have any pathogenic variants in their plasma or tumor sample (15.38%). A total of 11 (42.3%) patients had the same mutational profile in their plasma and tumor tissue. The most frequent mutations identified by NGS plasma and tumor analysis were those of TP53, followed by KRAS and EGFR gene (Figure 1). In addition to the variants detected in these genes, the following genes were found to have mutations upon NGS analysis: MET, PTEN, BRAF, NRAS, PIK3CA, FGFR1, FGFR3, FBXW7, SF3B1. 

The most frequent pathogenic variant was EGFR c.2573 T > G p.Leu858 Arg (L858R) with an allelic frequency ranging from 20.19 to 90.99% in tumors and 0.37% to 23.62% in plasma. A total of 23 patients were tested for EGFR mutations by real-time PCR to compare the results obtained from two different methods on the tissue samples. The same comparison was not accomplished for the liquid biopsy. Real-time PCR analysis was not performed due to the limited amount of plasma collected from the enrolled patients. The pathogenic variant L858R in exon 21 was detected in three patients (13.04%). The PCR results were confirmed by NGS analysis of the tumor tissue (3/26 or 11.5%). No other EGFR mutation was detected in the tumor tissue by PCR or NGS. Interestingly, two other patients exhibited this mutation in plasma, but not in the tumor tissue, neither by PCR nor by NGS analysis. The allelic frequency of this mutation was 0.26% and 0.37% respectively, in the two plasma samples.

When analyzing tumor tissue, 11 out of 26 or 42.3% of overall patients had at least one TP53 mutation with a slightly higher percentage for SSC patients—55%—compared with ADK patients—37.5%. Following plasma analysis, TP53 variants were detected in 13/26 (50%) patients. All variants found in the FFPE analysis were detected in the plasma of the same patient, except a single variant of uncertain significance with an allelic frequency of 5.22%, which was not detected in the plasma. For three of the patients with similar TP53 variants found in plasma and tumor, another additional different TP53 variant was found in plasma that did not have correspondence in the FFPE material of the same patient. One of these patients had a variant that is not included in the Oncomine Solid Tumor panel used for FFPE NGS analysis.

Four patients (15.4%) exhibited a KRAS variant both in their plasma and their tumor. One of these patients had an additional KRAS variant in the plasma with an allele frequency of 0.57% that was not found in the tumor. Another patient had a single KRAS mutation (variant G13D) that was not present in the FFPE.

The most frequent variant in our group was EGFR L858R, detected in 5 out of 26 (19.23%) plasma samples and in 3 out of 26 (11.53%) tumor samples, followed by the group of variants detected in codon 12 of KRAS gene (4 out of 26 or 15.38%) in plasma and tumor tissue.

### 3.3. Variant Allelic Frequency for Different Genes in Plasma and Tumor Tissue

With no exception, as expected, given other possible sources for the cfDNA in plasma, the mutation percent allelic variance was lower in plasma than detected in the tumor tissue (Wilcoxon signed-rank test, matched pairs, *p* < 0.0001 ***). Plasma VAF% ranged from 0.11% to 49.53%, while tumor tissue VAF% ranged from 1.40% to 49.53%. Almost all the mutations identified in the tumor could also be detected in plasma (detection rate of 25/26 or 96.15%), supporting the argument that, in advanced (stage III and IV) lung cancer patients, the liquid biopsy could be well qualified to take the place of the more invasive tissue biopsy to guide treatment (Figure 2). 

The values of allele frequency for the mutated genes detected in plasma correlated positively with their tumor counterparts, with mutations prevalent in the tumor being robustly detectable in plasma. Less expressed tumor mutations were still detectable in plasma, although some at values closer to the 0.1% allele frequency threshold (Spearman’s r = 0.6848, *p* < 0.0001, see Figure 3).

### 3.4. The Correlation between Risk Factors and Lung Cancer

The influence that exposure to toxic compounds for the respiratory system has on age of NSCLC diagnostic was evaluated (Figure 4B). The group of patients that worked in a toxic environment developed lung cancer earlier in life (median age of 65 years) compared with the group working in a clean environment (median age of 69.5 years, n.s., Unpaired *t*-Test). Smoking had a more aggressive influence on the age-correlated lung cancer diagnostic. The never smoked group had a much more advanced median age of NSCLC diagnostic compared with the ex-smoker and smoker group (76 years vs 70 and 64, respectively, never smoked vs. smoker groups *p* < 0.05 *, ANOVA, Dunnett multiple comparison test). A significant negative correlation was noticed when considering the cumulative influence of all toxic compounds that affected the respiratory system of our NSCLC patients during their life until the age of diagnosis.

Among the genes with pathogenic variants detected in the tissue samples, KRAS mutation seems to be associated with smoking, all four detected KRAS mutations being in current smokers (*p* = 0.098, n.s., Fisher’s Exact Test, Current Smokers vs. Never Smoked and Ex-Smokers). TP53 gene mutations were prevalent in ex-smoker and current smokers subgroups (*p* = 0.098, n.s., Fisher’s Exact Test, Current Smokers vs. Never Smoked and Ex-Smokers), while for EGFR, a higher prevalence was observed in non-smokers (*p* = 0.065, n.s., Fisher’s Exact Test, Never Smoked vs. Current Smokers). 

## 4. Discussion

Precision oncology approaches patients in a personalized manner based on their own tumor molecular profile, which can be investigated in our days by NGS assay. Regardless of their incidence, in many cancers, including NSCLC, tissue harvest for analysis purposes is an issue. Therefore, alternative methods analyzing tumor-derived genetic material, such as liquid biopsy, hold great potential in overcoming this disadvantage and opening personalization perspectives for these patients. In this context, the aim of the present study was to report our experience in using a dual approach of plasma and tumor NGS analysis for the benefit of patients diagnosed with NSCLC. To this end, a group of 26 eligible patients was selected and the presence of specific pathogenic variants was investigated in paired plasma and tumor tissue. Plasma NGS analysis revealed more pathogenic variants than tumor tissue investigations in our group of patients. There are studies presenting a different conclusion, with a lower sensitivity for plasma NGS compared with tumor tissue mutational status evaluation [13]. However, our study group focused only on advanced stages (III/IV) NSCLCs. This is a tumor characteristic found to contribute to increased concentrations of tumor cfDNA in plasma [14], facilitating the detection of a higher mutational heterogeneity. Nevertheless, the presence of false positive results in the plasma due to clonal hematopoiesis has been demonstrated. A percentage of 5–13% of people over the age of 70 people, or 25% of patients with cancer accumulate a greater number of mutations in their non-malignant hematopoietic cells and as a high percent of plasma cfDNA comes from these cells, the hematopoietic somatic mutations can be detected in plasma together with the tumor mutations [15,16].

From all 29 different mutations detectable by both NGS panels (plasma and tumor), seven different variants (24.13%; EGFR L858R in two patients, KRAS G13D and Q61H and TP53 G244D, V197M, R213P, and R273H) were detected only in plasma and not in the tumor itself. These mutations were detected in seven different patients, two of them having known distant organ metastasis, with an age of 58–77 years (median, 65). Thus, the source of the mutated cfDNA could be either the primary tumor site (avoiding detection in tissue due to sampling from heterogeneous tumors) or the metastatic site, which has evolved its own mutations. A larger study could shed more light on this matter and, if a connection between plasma mutations and metastatic tumors can be identified, it could provide a more informed therapeutic decision process for patients with advanced or metastatic lung cancer. 

Although current guidelines indicate that plasma variant detection be used only as a back-up choice for the cases where a tumor biopsy cannot be obtained [17], there have been more and more studies that confirm the efficiency of plasma NGS analysis, especially on metastatic tumors where it can reveal the heterogeneity of the patient disease with minimum invasion [18,19,20] and can provide real-time information on the efficiency of the targeted treatment [21,22].

The most frequent pathogenic variant in our study was EGFR L858R. This was the only mutation detected in EGFR gene in our patient group (5/26 (19.23%). It has been confirmed that L858R together with exon 19 deletions account for over 80% of all EGFR mutations in NSCLC. No deletion in EGFR exon 19 was detected in our study group, however, there are other studies that confirm L858R as being the most frequent mutation in the EGFR variants group [23,24,25].

PCR is considered to fail in detecting approximately 10% of EGFR mutations [26,27]. In our group, there was a 100% reproducibility between PCR-detected EGFR variants and tumor tissue NGS results, and a 92.3% concordance between tumor tissue PCR and NGS and plasma NGS results, similar to other studies [28,29].

The most common genetic alterations associated with cancers are pathogenic variants in the TP53 gene [30]. Gene mutations in TP53 are predominantly associated with EGFR mutations [31] and KRAS mutations [32,33] in NSCLC patients and are indicators for unfavorable progression-free and overall survival [34]. In our study group, TP53 was the most frequent mutated gene. Only one patient had mutations of this suppressor gene in co-occurrence with EGFR L858R, on the other hand, there were two cases with KRAS and TP53 co-occurring mutations.

More than 80% of KRAS alterations in NSCLC are detected in codon 12, with almost 40% of the KRAS mutated cases harboring G12C variant [35]. Our results revealed a slightly lower prevalence of G12C variant (33.3% of the total number of KRAS variants), and a lower percentage of mutations found in codon 12 (66.7%). 

All KRAS and EGFR mutations were observed in patients with lung adenocarcinoma (5/26 patients each, representing 31.25%) in accordance with previously published data showing KRAS mutations in 18–32% of ADKs and only 1.6–7.1% of SSCs [36] and EGFR mutations in about one-third of ADKs [37,38] and in 3–18% of SSCs [39].

Smoking is an important mutagen and its defining role in lung cancer is no longer disputed. We found that KRAS and TP53 were the most affected DNA targets of cigarette smoke damage, while EGFR mutations were present with a higher rate in non-smokers versus smokers. Mutations in the TP53 gene have previously been found to be associated with environmental exposure [40]. Moreover, there are numerous studies concluding that TP53 mutations were predominantly found in smokers, with a higher frequency than in non-smokers [30,41,42,43] and that EGFR alterations are rather associated with non-smoker status rather than induced by cigarette smoke [29].

An already known acquired mutation is T790, used in clinical practice, and frequently performed from liquid biopsy due to the impossibility of harvesting tissue from all metastatic lesions [44,45]. Consequently, there were different attempts to include liquid biopsy in the tumor types but for the moment, only lung and colon cancer are the winners.

Any kind of malignancy is the result of somatic accumulation of mutations that empower the malignant cells with the ability to survive and proliferate without control. NSCLC is characterized by a high rate of mutations and therefore, NGS has proven very useful in this localization. 

In the metastatic setting, when the biopsy is obtained through minimally invasive techniques (e.g., bronchoscopy), the tumoral tissue is limited, and therefore, it is difficult to assess all the bronchopulmonary cancers mutations. Moreover, NGS analysis is superior to immunohistochemistry and PCR assay, as it has a higher specificity and sensitivity and can provide predictive information about the clinical response to targeted therapies. However, the availability of tumor tissue for NGS is a problem in many cases, especially due to depletion and pre-analytical damaging of the tissue or the impossibility to collect a biopsy from the patient. Testing the cfDNA from liquid biopsy is becoming an important additional tool that could bring valuable information on the tumor molecular characteristics and guide targeted therapy decision [46]. The number of liquid biopsy assays that are validated for different targeted treatments is growing [47,48,49]. There are many advantages to analyzing cfDNA, such as faster results due to the easier processing of blood samples compared to tumor tissue; furthermore, the procedure of obtaining a liquid biopsy from the patient is much less invasive [11]. Nevertheless, the percent of circulating cell-free tumor DNA (ctDNA) in plasma can be very low compared to the total amount of existing cfDNA reducing the sensitivity of the results [50]. Special attention needs to be given to the rapid processing of blood after the collection of samples or the usage of special preservative collection tubes, due to the fact that the half-life of ctDNA is very short [10]. However, serial analysis of the cfDNA from liquid biopsy can provide valuable information on the clonal evolution and heterogeneity of NSCLC, with an early detection of the therapy resistant mechanisms developed by the disease [51]. As a result of all the advantages and disadvantages that cfDNA testing brings, liquid biopsy analysis is considered for the moment a valuable complementary investigation in advanced-stage NSCLC patients. In some cases where the availability of tumor tissue is limited, the investigation of concomitant cfDNA and tumor sample is adopted [52]. However, the rapidity of sample processing and the facility with which the liquid biopsy is obtained determine in many cases the “blood first” approach [53]. Nevertheless, a lack of pathogenic variant obtained from a plasma sample should be considered cautiously and confirmed with a screening of the tumor tissue whenever the availability of the sample allows it. 

Apart from its utility in advanced-stage NSCLCs, there are ongoing studies exploring the feasibility and utility of liquid biopsy in early-stage NSCLC regarding driver mutations [54]. Some preclinical studies have been conducted on PD-L1 expression on circulating tumor cells (CTC) for addressing them with immune checkpoint inhibitors. Tumor mutation burden (TMB) has been explored in liquid biopsy for identifying the target for immunotherapy, but with inconstant results [55]. Liquid biopsy is expected to gain an important role in the routine diagnostic of early-stage NSCLC, before any clinical or imagistic sign of disease, with impact on earlier treatment and improved long-term survival [54]. 

Clinical implications of liquid biopsy could consist of prognosis, identification of therapeutic targets and resistance, monitoring of target therapy, real-time monitoring of disease, and identifying the best patients for an immunotherapy approach. 

## 5. Conclusions

Liquid biopsy and tissue biopsy hold their own strengths. On the one hand, tissue biopsy is still the main criterion for diagnosis in cancer, while liquid biopsy offers a huge advantage of non-invasive real-time molecular analysis of the disease status. Despite the limited data obtained in our study due to the small number of NSCLC patients enrolled, we can conclude that plasma cfDNA testing using a panel of targeted genes could be successfully used as a complementary analysis for molecular testing in patients with NSCLC. Our data show that NGS analysis of cfDNA could identify actionable mutations in advanced NSCLC and therefore, this analysis could be successfully used to monitor the disease progression, the treatment response and even to modulate the therapy if needed.

The main limitation of the study is the small number of patients and related to that, the limitation to NSCLC cases. In the near future, one goal is to gather and analyze a bigger number of patients with NSCLC. Another aim is to obtain FFPE and liquid biopsy samples and to analyze them from at least 30–40 cases of SCLC, considering its different biological behavior, management, and unfavorable prognostic.

## Figures and Tables

**Figure 1 cancers-14-06084-f001:**
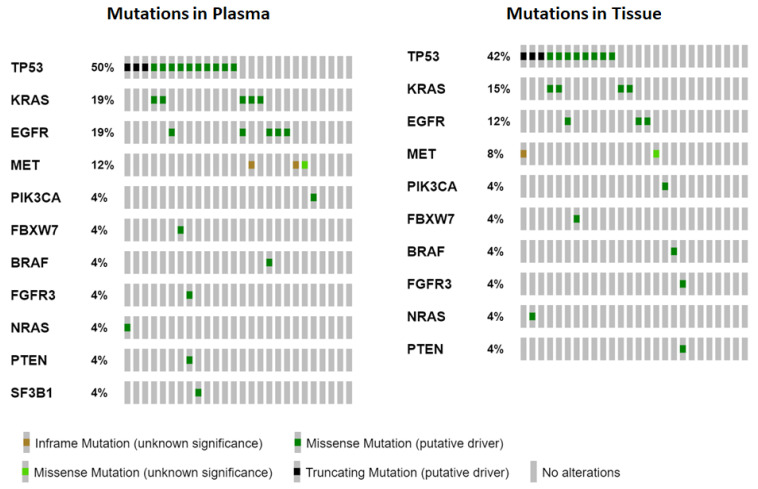
OncoPrint comparative illustrations showing frequencies and types of alterations for the identified genes (rows, sorted by percent alterations) in the study patients (columns) for both plasma samples (left) and tissue samples (right) (https://www.cbioportal.org/oncoprinter (accessed on 5 November 2022)).

**Figure 2 cancers-14-06084-f002:**
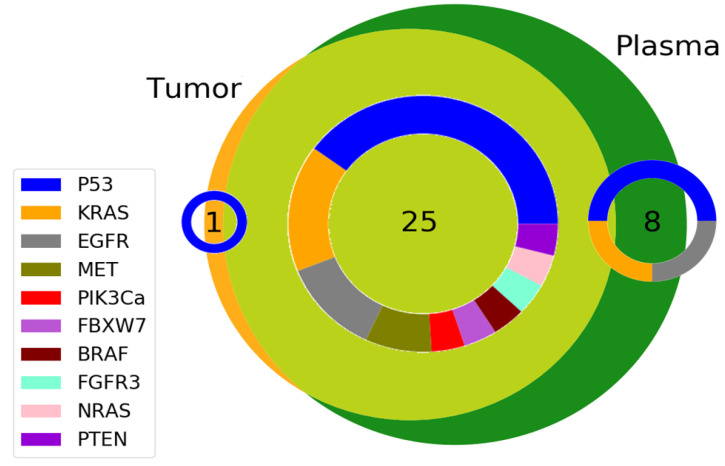
Venn diagram presenting (i) the number of mutations identified both in plasma and tumor tissue, (ii) the number of mutations in tumor tissue only, and (iii) the number of mutations in plasma only. The donut diagram shown over the number values illustrates the genes composing that number.

**Figure 3 cancers-14-06084-f003:**
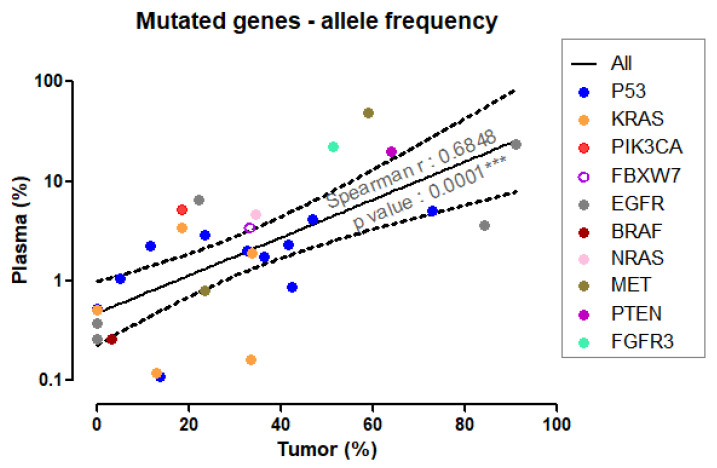
Correlation between plasma and tumor allele frequency. As plasma allele frequency followed a log-normal distribution, these data were log10-normalized, while the tumor values, passing the Kolmogorov–Smirnov normality check, were used as a percentage. The solid line represents the linear regression best fit, dashed lines show the 95% confidence interval. The slope deviated significantly from zero, with a *p*-value < 0.0001.

**Figure 4 cancers-14-06084-f004:**
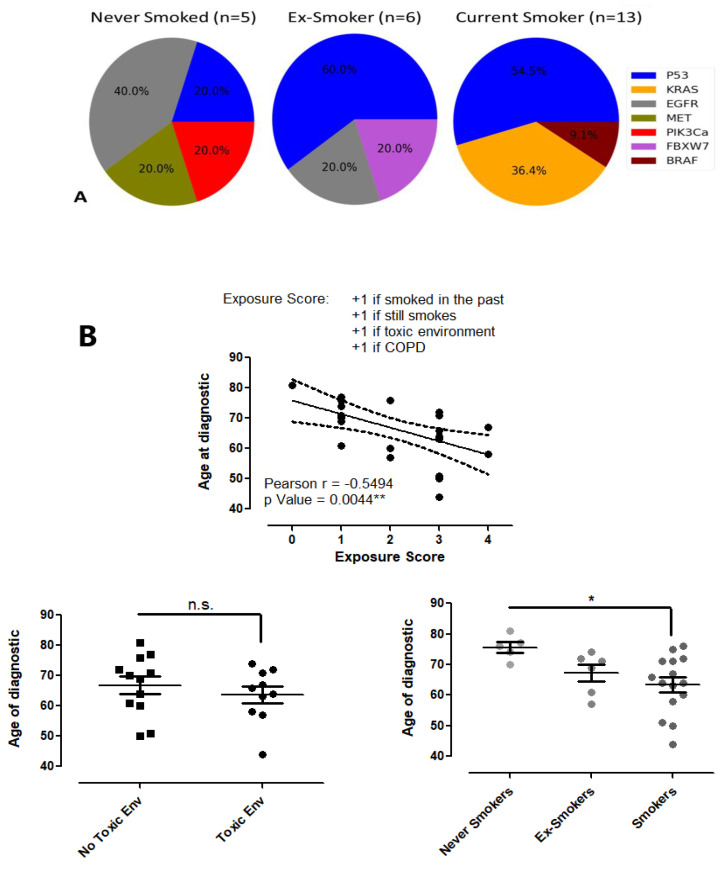
Lung cancer and risk factors. (Panel **A**). Pie charts of tissue mutations for lung cancer patients grouped by their smoking status. (Panel **B**). The age at lung cancer onset for patients with or without professional exposure to toxic substances (n.s., Student’s *t*-test), for non-smokers, ex-smokers, and current smokers (* *p* < 0.05, Current Smokers vs. Never Smoked, one-way ANOVA, Dunnett multiple comparison test), as well as the significant negative correlation between a compound exposure score and age at lung cancer onset (** *p* < 0.01, Pearson correlation).

**Table 1 cancers-14-06084-t001:** IHC antibodies.

No	Antigen	Clone	Provider
1	TTF1	SP141	Ventana Medical Systems, Tucson, AZ, USA
2	CK7	SP52	
3	KI67	30-9	
4	Synaptophysin	SP11	
5	Vimentin	v9	
6	CD5	SP19	
7	PD-L1	SP263	
8	ALK	D5F3	
9	Calretinin	SP65	
10	CK20	SP33	
11	PAX8	MRQ-50	
12	P63	7JUL	Leica Biosystems, Deer Park, IL, USA
13	Chromogranin A	5H7	
14	Wilms’ Tumor	WT49	
15	CDX2	EP25	
16	CA19.9	Syalyl Lewis C241:5:1:4	
17	ER	6F11	
18	GATA3	L50-823	Bio SB, Santa Barbara, CA, USA
19	P40	ZR8	
20	PD-L1	22C3	Agilent Technologies, Santa Clara, CA, USA

**Table 2 cancers-14-06084-t002:** Description of NSCLC patients’ analysis group.

	Never Smoked	Ex-Smokers	Current Smokers	All Patients
**Number**	5	6	13	26
**Age (mean ± SD)**	76 ± 3.6 years	67 ± 6.2 years	62 ± 8.9 years	67 ± 9 years
**Sex**				
Female	3 (60%)	0 (0%)	2 (15.4%)	6 (23.08%)
Male	2 (40%)	6 (100%)	11 (84.6%)	20 (76.92%)
**Differentiation degree**				
Well differentiated (G1 and G1/G2)	0 (0%)	0 (0%)	0 (0%)	0 (0%)
Moderately differentiated (G2)	1 (20%)	1 (16.7%)	2 (15.4%)	4 (15.38%)
Poorly differentiated ( G3)	2 (40%)	4 (66.7%)	10 (76.9%)	18 (69.23%)
NA	2 (40%)	1 (16.7%)	1 (7.7%)	4 (15.38%)
**TNM stage**				
Stage I/II	0 (0%)	0 (0%)	0 (0%)	0 (0%)
Stage III/IV	5 (100%)	6 (100%)	13 (100%)	26 (100%)

**Table 3 cancers-14-06084-t003:** Variants detected by NGS analysis in paired tumor and plasma samples.

PATIENT No.	GENE	MUTATION	VAF % TUMOR	VAF% PLASMA
1	MET	T1010I	58.89	49.53
2	MET KRAS	c.3082 + 2T > C p.? G13D	- -	0.81 0.51
3	MET TP53	exon 14 Skipping c.2942-11_2986del p.? R213 * V143M	- 23.31 1.40 -	- - 0.80 0.65
4	SF3B1 TP53	K700E G244D	- -	0.25 0.52
5	FBXW7 TP53	R505S R248W	33.1 73.01	3.46 5.07
6	-	-	-	-
7	PIK3CA	E545K	18.3	5.28
8	TP53	[E271D; V272L] V272L	5.08 5.22	1.07 -
9	TP53	R282G	46.79	4.14
10	FGFR3 PTEN TP53	R248C D24Y C141W	51.35 64.07 -	22.61 20 16.24
11	TP53	C242Y	36.3	1.74
12	EGFR TP53	L858R R248W V197M	90.99 41.62 -	23.62 2.33 0.26
13	-	-	-	-
14	KRAS TP53	G12C G245S R213P	33.77 42.3 -	1.89 0.88 1.26
15	-	-	-	-
16	TP53	S303Afs*42	32.71	2.02
17	EGFR	L858R	22.19	6.52
18	EGFR	L858R	84.25	3.61
19	BRAF EGFR	V600E L858R	3.06 -	0.26 0.26
20	KRAS	G12C	33.46	0.16
21	KRAS TP53	G12V R213L	18.36 11.61	3.40 2.23
22	KRAS EGFR	G12D Q61H L858R	12.81 - -	0.12 0.57 0.37
23	NRAS TP53	Q61L G266 * R273H	34.53 23.48 -	4.72 2.89 0.82
24	-	-	-	-
25	TP53	S215I	13.69	0.11
26	MET	c.3082 + 1 G > C p.?	-	11.68

*—stop codon.

## Data Availability

The data are contained within the article.

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
