# Peer review of "Dual NGS Comparative Analysis of Liquid Biopsy (LB) and Formalin-Fixed Paraffin-Embedded (FFPE) Samples of Non-Small Cell Lung Carcinoma (NSCLC)"

_cancers, 2022, doi:10.3390/cancers14246084_

Round 1

Reviewer 1 Report

The manuscript presented by Laura Buburuzan et al. shows some premises, however, its general message does not present sufficient value to be published in its current form. The main shortcomings of the study are the wrong-defined aim of the study, a too small studied group for evaluating statistical significance, no information about the time point when the LBs samples were collected, and the clinical follow-up of the cohort. The article requires sequential review before acceptance.

Major remarks

1. The introduction only limits the epidemiological aspects of NSCLC, which is not interesting for the reader itself while the main topic of the paper considers both application of NGS and liquid biopsy in NSCLC. Thus these topics should be elaborated on in detail in the introduction. The first paragraph of the discussion may be incorporated into the intro, however, it requires to be more detailed and updated by the new reports.

2. The aim of the study is wrongly defined at the end of the introduction. The Oncomine PanCancer Cell-Free Assay was already well-established and validated by ThermoFisher. The aim of the study seems to be to distinguish differences in the genetic landscape in FFPE and liquid biopsy samples.

3. Please indicate the time point when the liquid biopsy was sampled in the studied group. The different time points of sampling may affect the genetic landscape in liquid biopsy.  

4. To visualize NGS data on all figures I suggest applying the OncoPrint R package.

5. Could authors define well the aim of application PCR genotyping of FFPE samples? It seems to be the validation of the NGS results. thus should be reported after them in the Methods description. To provide the full validation the liquid biopsy samples should be validated by PCR too.

6. Table 1 should be implemented by the smoking status.

7. There is no info about the treatment applied in the studied patents and no data about their follow-up. The results were not correlated and discussed with the clinical outcome of the studied cohort.

8. The analysis of metastatic samples would be informative to verify the clonal history of mutated cfDNA.

9. The wide discussion about future perspectives of liquid biopsy application NSCLC is required.

Minor remarks:

1. Abstract should reflect only the number of cases finally sequenced. Only 26 patients were studied, thus reporting that 57 patients were included in the study leads to misunderstanding for the reader.

2. The antibodies used in the IHC studies should be simply summarized in the table rather than described in the text.

3. The TCGA abbreviations for lung adenocarcinoma and lung squamous cell carcinoma are LUAD and LUSC respectively

4. Lane 211 – typo in the word “FFPET” that is used 3 times in the text.

5. Authors in the text use cDNA instead of cfDNA which is wrong.

6. Lanes 393 -410 require references.

Author Response

Dear reviewer,

Thank you for reviewing our work. We consider that the quality of our manuscript was increased after responding to your comments. Please find below our answers and the modifications in the new upload.

Major remarks

Q1. The introduction only limits the epidemiological aspects of NSCLC, which is not interesting for the reader itself while the main topic of the paper considers both application of NGS and liquid biopsy in NSCLC. Thus these topics should be elaborated on in detail in the introduction. The first paragraph of the discussion may be incorporated into the intro, however, it requires to be more detailed and updated by the new reports.

A1. The first paragraph from the Discussions section was moved into the introduction as suggested. The following paragraph was added to the already existing introduction:

“NGS analysis has also the advantage of uncovering some unexpected gene variants that can benefit from targeted therapy (De Maglio et al., 2022). However, in particular cases of patients with worsening conditions when a fast result is needed, single-gene analysis by real-time PCR is a suitable option despite of a more restrictive information provided. A major disadvantage of the NGS analysis is the increased turnaround time, that often exceeds in many labs the 10 days stipulated in the international guidelines (Lindeman et al., 2018). Despite the obvious informative advantage of tumor tissue NGS, the analyst encounters frequently the impediment of the lack of tissue following immunohistochemistry investigation, or the pre-analytical damage of the tumor tissue. These types of situations make the NGS analysis of cell-free tumor DNA isolated from liquid-biopsy a precious instrument for molecular investigations. Liquid biopsy analysis is an important tool for patients with advanced stages of NSCLC or in the situations where it is not possible to obtain a biopsy (Rolfo et al., 2018) and is recently considered as a complementary investigation to tissue-based approach (Tan, 2022).”

Q2. The aim of the study is wrongly defined at the end of the introduction. The Oncomine PanCancer Cell-Free Assay was already well-established and validated by ThermoFisher. The aim of the study seems to be to distinguish differences in the genetic landscape in FFPE and liquid biopsy samples

A2. The aim of the study was rephrased. However, the Oncomine PanCancer Cell-Free Assay is a wide panel targeting multiple genes and we aimed to validate its use in particular for NSCLC. Please find below the new paragraph:

“The main aim of the current study was to distinguish the potential differences between the molecular landscapes found in the tumor tissue and in plasma samples harvested from patients with NSCLC by NGS. As a result, we validated the potential use of the Ion Torrent™ platform and technology for NGS of cfNAs in NSCLC using Oncomine™ Pan-Cancer Cell-Free Assay as a valuable tool for prospective NSCLC monitoring and therapy modulation”.

Q3. Please indicate the time point when the liquid biopsy was sampled in the studied group. The different time points of sampling may affect the genetic landscape in liquid biopsy.  

A3. The peripheral blood samples were harvested in the same time with the tumor tissue. The word “simultaneously” was added in paragraph 2.1 Pateients to clarify this aspect.

Q4. To visualize NGS data on all figures I suggest applying the OncoPrint R package.

A4. Figure 1 was replaced by an illustration obtained using the Oncoprint online (https://www.cbioportal.org/oncoprinter). The figure caption as well as section 2.7. Data analysis were modified accordingly.

Q5. Could authors define well the aim of application PCR genotyping of FFPE samples? It seems to be the validation of the NGS results. thus should be reported after them in the Methods description. To provide the full validation the liquid biopsy samples should be validated by PCR too.

A5. The aim of PCR application in FFPE samples was stated in the original manuscript under Table 2 (row 257-259): “A number of 23 patients were tested for EGFR mutations by real-time PCR to compare the results obtained from two different methods on the tissue samples”.

Paragraph: 2.6. Tumor tissue and PCR evaluation of EGFR pathogenic variants was moved to the end of the Materials and methods section as suggested.

We agree that the full validation would have been done on liquid biopsy as well, but due to the limited amount of plasma, this was not accomplished. We added a clear statement in the manuscript rows 259-261: “The same comparison was not accomplished for the liquid biopsy. Real-time PCR analysis was not performed due to the limited amount of plasma collected from the enrolled patients

Q6. Table 1 should be implemented by the smoking status.

A6. Table 2 (original Table 1) was modified accordingly

Q7. There is no info about the treatment applied in the studied patents and no data about their follow-up. The results were not correlated and discussed with the clinical outcome of the studied cohort.

A7. Considering that this is a one time point experiment, with samples harvested before the treatment, there is no impact of the treatment on the data analysis. Therefore, we didn’t collect or considered such information for this study.

Q8. The analysis of metastatic samples would be informative to verify the clonal history of mutated cfDNA.

A8. We totally agree with your point. However, the tumor tissues used to come from biopsies performed in many cases on patients without known metastases at that time. No access to metastatic tissues was available, therefor we limited our experiment to comparing the molecular landscape found in primary tumor biopsies and plasma.

Q9. The wide discussion about future perspectives of liquid biopsy application NSCLC is required.

A9. In the Discussion section, the first paragraph was moved into the Introduction and a wide discussion was added. Please see the inserted text below:

However, the availability of tumor tissue for NGS is a problem in many cases, especially due to depletion and pre-analytical damaging of the tissue or the impossibility to collect a biopsy from the patient. Testing the cfDNA from liquid biopsy is becoming an important additional tool that could bring valuable information on the tumor molecular characteristics and guide targeted therapy decision (De Luca, 2020). The number of liquid biopsy assays that are validated for different targeted treatments is growing (Wolf ae al, 2020, Paik et al, 2020, Bauml et al., 2021). There are many advantages in analyzing cfDNA, such as faster results due to the easier processing of blood samples compared to tumor tissue, also, the procedure of obtaining a liquid biopsy from the patient is much less invasive (Tan, 2022). Nevertheless, the percent of circulating cell-free tumor DNA (ctDNA) in plasma can be very low compared to the total amount of existing cfDNA reducing the sensitivity of the results (Bettegowda et al., 2014). Special attention needs to be given to the rapid processing of blood after the collection of samples or the usage of special preservative collection tubes, due to the fact that the half-life of ctDNA is very short (Rolfo et al., 2018). However, serial analysis of the cfDNA from liquid biopsy can bring valuable information on the clonal evolution and heterogeneity of NSCLC, with an early detection of the therapy resistant mechanisms developed by the disease (Fernandes et al., 2021). Due to all the advantages and disadvantages that cfDNA testing brings, liquid biopsy analysis is considered for the moment a valuable complementary investigation in advanced stage NSCLC patients. In some cases, where the availability of tumor tissue is limited the investigation of concomitant cfDNA and tumor sample is adopted (Aggarwal et al., 2021). However, the rapidity of sample processing and the facility with which the liquid biopsy is obtained, determined in many cases the “blood first” approach (Leighl et al., 2019). Nevertheless, a no pathogenic variant obtained from a plasma sample should be considered cautiously and confirmed with a screening of the tumor tissue whenever the availability of the sample allows it.

Apart from its utility in advanced stage NSCLCs, there are ongoing studies exploring the feasibility and utility of liquid biopsy in NSCLC early stage regarding driver mutations (X). Some preclinical studies were conducted on PD-L1 expression on circulating tumor cells (CTC) for addressing them with immune checkpoint inhibitors. Tumor mutation burden (TMB) is explored in liquid biopsy for identifying the target for immunotherapy, but with inconstant results (Y). Liquid biopsy is waited to gain an important role in routine diagnostic of more early stage NSCLC, before any clinical or imagistic sign of disease, for an earlier treatment and improved long term survival (X) ”.

More, a short paragraph was also added in the Conclusions section:

The main limitation of the study is the small number of patients and related to that, the limitation to NSCLC cases. In the near future, one goal is to gather and analyze a bigger number of patients with NSCLC. Another desiderate is to obtain FFPE and liquid biopsy samples and to analyze them from at least 30-40 cases of SCLC, considering its different biological behavior, management and unfavorable prognostic

Minor remarks:

Q1. Abstract should reflect only the number of cases finally sequenced. Only 26 patients were studied, thus reporting that 57 patients were included in the study leads to misunderstanding for the reader.

A1. The statement was eliminated from the abstract to avoid misunderstandings.

Q2. The antibodies used in the IHC studies should be simply summarized in the table rather than described in the text.

A2. The antibodies used in the IHC studies were summarized in table 1

No

Antigen

Clone

Provider

1

TTF1

SP141

Ventana Medical Systems, Tucson, AZ, USA

2

CK7

SP52

3

KI67

30-9

4

Synaptophysin

SP11

5

Vimentin

v9

6

CD5

SP19

7

PD-L1

SP263

8

ALK

D5F3

9

Calretinin

SP65

10

CK20

SP33

11

PAX8

MRQ-50

12

P63

7JUL

Leica Biosystems, Deer Park, IL, USA

13

Chromogranin A

5H7

14

Wilms' Tumor

WT49

15

CDX2

EP25

16

CA19.9

Syalyl Lewis C241:5:1:4

17

ER

6F11

18

GATA3

L50-823

Bio SB, Santa Barbara, CA, USA

19

P40

ZR8

20

PD-L1

22C3

Agilent Technologies, Santa Clara, CA, USA

Q3. The TCGA abbreviations for lung adenocarcinoma and lung squamous cell carcinoma are LUAD and LUSC respectively

A3. The abbreviations were introduced in the text where appropriate.

Q4. Lane 211 – typo in the word “FFPET” that is used 3 times in the text.

A4. We corrected all the typos.

Q5. Authors in the text use cDNA instead of cfDNA which is wrong.

A5. This typo was corrected.

Q6. Lanes 393 - 410 require references.

A6. A reference was added.

Reviewer 2 Report

The authors describe future technologies in diagnostics of cancer.  Minimally invasive Liquid Biopsy can replace the standart fomalin-fixed biopsy. 

Minor comments: 

Line 46 - need the references 

Line 49 - need the references 

Line 43 - give the full name of GLOBOCAN

Line 199 - write the full information about EGFR variant ( chromosomal change, coding and protein)

Line 314  - need the references 

In discussion, please write Paragraph of disadvantages of  Liquid Biopsy and using it in real clinical practice. 

Question: Why you have chosen only NSCLC from lung cancer? What was the meaning? 

Author Response

Dear reviewer,

Thank you for your comments. Please find below our responses.

Minor comments: 

Q1. Line 46 - need the references 

A1. Thank you for your comment, a reference was added.

Q2. Line 49 - need the references 

A2. Thank you for your comment, a reference was added.

Q3. Line 43 - give the full name of GLOBOCAN

A3. Thank you for your comment, the full name was stated before the abbreviation.

Q4. Line 199 - write the full information about EGFR variant (chromosomal change, coding and protein)

A4. Thank you for your comment, the requested information was added in the text.

Q5. Line 314 - need the references 

A5. Thank you for your comment, a reference was added.

Q6. In discussion, please write Paragraph of disadvantages of Liquid Biopsy and using it in real clinical practice. 

A6. Thank you for your comment, the discussions section was enlarged. A paragraph on the Liquid Biopsy limitations was introduced in the Conclusion section.

Q7. Question: Why you have chosen only NSCLC from lung cancer? What was the meaning?

A7. Thank you for your comment, a clear statement regarding the motivation of the NSCLC study was added in the introduction.

Round 2

Reviewer 1 Report

The article was improved by the first major revision in the easiest points. However, it still does not represent a high quality to be published in Cancers.